# Agricultural pesticide use and adverse birth outcomes in the San Joaquin Valley of California

Ashley E. Larsen[1], Steven D. Gaines[1] & Olivier Deschênes[2]

Virtually all agricultural communities worldwide are exposed to agricultural pesticides. Yet, the health consequences of such exposure are poorly understood, and the scientific literature remains ambiguous. Using individual birth and demographic characteristics for over 500 000 birth observations between 1997–2011 in the agriculturally dominated San Joaquin Valley, California, we statistically investigate if residential agricultural pesticide exposure during gestation, by trimester, and by toxicity influences birth weight, gestational length, or birth abnormalities. Overall, our analysis indicates that agricultural pesticide exposure increases adverse birth outcomes by 5–9%, but only among the population exposed to very high quantities of pesticides (e.g., top 5th percentile, i.e., ~4200 kg applied over gestation). Thus, policies and interventions targeting the extreme right tail of the pesticide distribution near human habitation could largely eliminate the adverse birth outcomes associated with agricultural pesticide exposure documented in this study.

[1] Bren School of Environmental Science & Management, University of California, Santa Barbara 93106-5131, USA. [2] Economics Department, University of California, Santa Barbara 93106-9210, USA. Correspondence and requests for materials should be addressed to A.E.L. (email: Larsen@bren.ucsb.edu)

For millennia, agriculture has had far-reaching impacts on human society and natural systems[1–3]. However, the development of modern pesticides, alongside other technological advances of the green revolution, caused dramatic production increases but also concomitant increases in environmental and health concerns. Numerous studies have documented negative effects of pesticides on a wide range of organisms as well as ecosystem services such as water and air quality[4, 5]. However, documentation of direct negative effects of non-occupational pesticide exposure on human health has proven much more elusive, despite substantial public apprehension[6, 7].

Reproductive harm tied to chemical exposure is of particular concern, because health at birth is correlated with both adult health[8] and non-health attributes (e.g., wages and educational attainment[9, 10]). Further, negative birth outcomes, such as low birth weight, preterm birth, and birth abnormalities, have been causally associated with other environmental conditions during pregnancy, including air pollution[11–13], extreme heat[14, 15], and maternal residence in proximity to toxic release sites[16]. Nevertheless, evidence linking residential agricultural pesticide exposure with adverse birth outcomes remains equivocal[7].

The absence of conclusive evidence of the health impacts of agricultural pesticide use may be due in part to the logistical challenges of health research. Since controlled studies are clearly unethical, much of the available evidence relating pesticides to adverse health outcomes comes from occupationally exposed groups, such as certified pesticide applicators[17–19], which may not reflect exposures that are relevant for the broader agricultural community. Due to a lack of refined pesticide use data for most regions, studies of non-occupational pesticide exposure either use broad, correlative measures of pesticide use at large scales or seek detailed measures of individual exposure via blood/urine samples or interviews. While such refined measures provide a valuable metric of exposure at multiple snapshots during gestation, the costs and logistical challenges associated with such approaches often constrain sample sizes to between 100–2000 births making statistical analyses of rare outcomes difficult[7, 17, 20, 21].

These logistical challenges have resulted in creative attempts to tie proxies of pesticide exposure to adverse birth outcomes. For instance, at the broadest scale, Winchester et al.[22] associated seasonal differences in pesticide concentrations in surface waters with national rates of birth defects. Similarly, Schreinemachers[23] investigated birth defects as a function of county wheat acreage—which they used as a proxy for herbicide exposure, while Petit et al.[24] investigated birth weight and fetal growth as a function of crop composition—which they interpret as a proxy for insecticides. These studies suggested significant negative effects of pesticides on birth defects[22, 23] and fetal growth[24]. However, studies with more refined measures of agricultural pesticide exposure and/or birth outcomes have generally reported null or inconsistent effects of exposure on birth defects[7, 20], low birth weight[25, 26], and gestational length[17], in part due to the consistently low number of comparisons[17, 26].

Previous studies using blood serum, urine samples, or interviews have often focused on specific chemicals or classes of chemicals[27, 28] for feasibility. Yet, in most agricultural communities a great diversity of chemicals are applied daily, making it difficult to isolate the effects of any specific toxic agent due to other chemicals in the environment and the understudied synergistic or antagonistic interactions among them[29, 30]. Even if one could isolate exposure to a specific chemical group, pesticide metabolites in blood serum and urine still reflect exposure to different chemicals across a range of toxicities within a given chemical group[31]. Further, half-lives range from a few hours to weeks for some chemicals of high concern, such as organophosphorous (OP) insecticides, making it difficult to

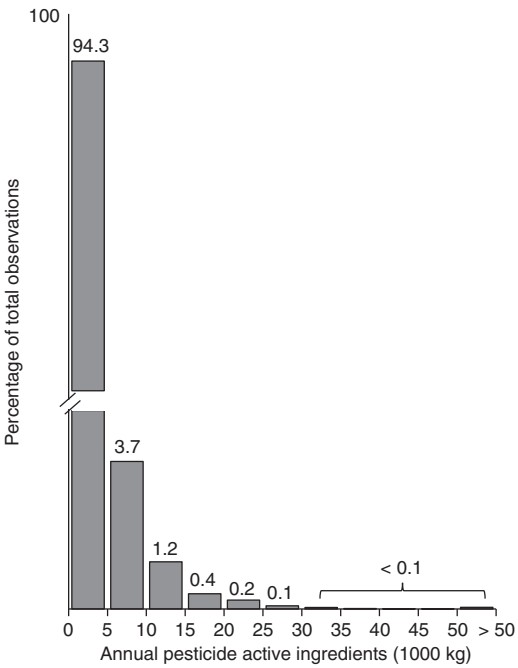

**Fig. 1** Distribution of annual pesticide active ingredients per PLS Section. Total annual pesticide active ingredients weighted by number of births follows a heavily skewed distribution, with almost 95% of observations experiencing less than 5000 kg per year, and the extreme right tail extending to substantially larger exposures. The y-axis is broken to illustrate the extremely small fraction of the population in the extreme right tail of the pesticide distribution

accurately capture exposure to even one chemical group over gestation.

The confounding effects of unobserved exposures or other unobserved factors (e.g., unobserved health behaviors like prenatal doctor visits) may underlie some of the counterintuitive results in the literature. For example, in an innovative pooled cohort study evaluating pesticide exposure and birth outcomes in an agricultural community in the Salinas Valley, CA as well as urban populations in New York and Cincinnati, Harley et al.[31] reported no significant effects of OP metabolite levels on birth weight overall or for the agricultural population in particular. Yet, the same study reported a negative effect of OP metabolite levels in urban minority populations, despite their lower average OP metabolite levels relative to the agricultural group. Additionally, studies have found statistically significant negative associations between living in proximity to agriculture and adverse outcomes (e.g., time to pregnancy), but not with pesticide metabolite levels directly[32]. Similarly counterintuitive results have illustrated that specific chemicals such as methyl bromide or OP pesticides have negative associations with some birth outcomes, but also unexpected positive associations (exposure increasing body length or gestational age) for others[30, 33].

Data-driven statistical approaches can provide complementary insight into these questions. Large samples provide a powerful opportunity to control for various different demographic and environmental characteristics that may be obscuring the relationship between agricultural pesticide exposure and adverse birth outcomes in surrounding communities.

Here we revisit the relationship between pesticide exposure and birth outcomes using a large sample of births (>500 000), which includes individual-level data on maternal and birth characteristics, and pesticide exposure at a small geographical scale. We concentrate on the agriculturally dominated San Joaquin

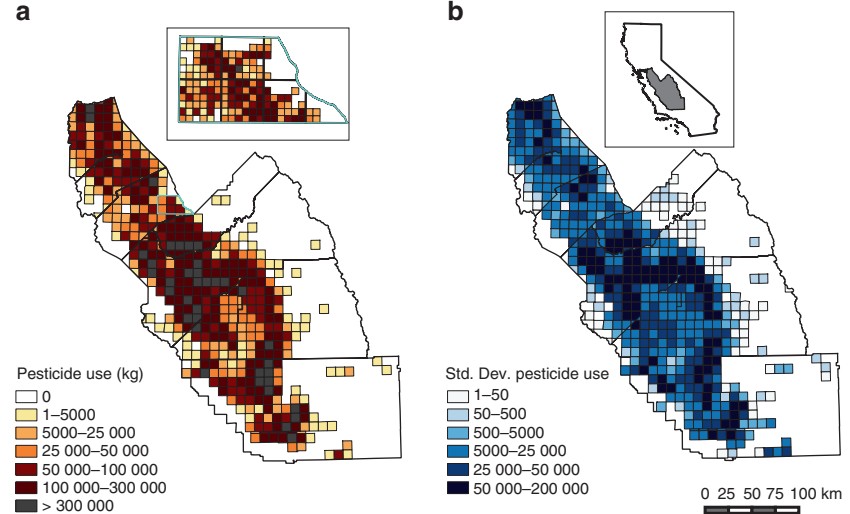

**Fig. 2** Spatial distribution of pesticide use and variability in California's San Joaquin Valley. The total kg of annual pesticide active ingredients varies over space and time as illustrated by the distribution of total kg by ~93 km² PLS Township in 2011 **a**, and the standard deviation of total kg by PLS Township **b**. The inlay on the left shows pesticide use at the 2.6 km² PLS Section, the unit of pesticide observation. The color scheme for the inlay corresponds to that of the PLS Township legend, adjusted for the difference in area (i.e., the most exposed category for the PLS Section is >8333 kg, 1/36th the values for the corresponding PLS Township category). The inlay on the right illustrates the study region within California. The *scale bar* corresponds to 100 km for the main panels

Valley, California. California is the most populous state in the United States with roughly 12% of annual births. It is also the greatest user of pesticides with over 85 million kg applied annually, an amount equivalent to roughly 30% of the cumulative active ingredients applied to US agriculture[34]. The San Joaquin Valley is the state's most productive agricultural region, growing an abundance of high value, high chemical input, and labor-intensive fruit, vegetable, and nut crops[35]. We evaluate pesticide exposure by summing active ingredients of agricultural pesticides applied over gestation, by trimester, and by grouped the United States Environmental Protection Agency's (EPA) acute toxicity categories, along with several additional robustness checks. For outcomes, we focus on birth weight, gestational age, and birth abnormalities. Our sample of over 500 000 individual birth observations and fine-scale data on the timing and amount of pesticide applied allow us to detect statistically significant negative effects of pesticide exposure for all birth outcomes, but generally only for pregnancies exposed to the very highest levels of pesticides (top 5 or 1% of exposure quantities).

## Results

**Summary statistics**. Pesticide use in the San Joaquin Valley follows an extremely skewed distribution (Fig. 1), and is highly variable year-to-year (Fig. 2). Average annual (12-month, January–December) pesticide use, weighted by number of births, was ~975 kg per ~2.6 km² area for 1997–2011. Not all areas in the San Joaquin Valley are agriculturally producing regions, and population centers with the most births have less agricultural land area. In our sample of singleton births to mothers residing in California (see Methods for other sample restrictions), over 50% of the births experienced zero pesticide exposure during a nine month gestation, where exposure is measured as kg of active ingredients applied in the 2.6 km² Public Land Survey Section ("PLS Section") encompassing the mother's address (Supplementary Table 1). Despite a median exposure of zero, average exposure over gestation was almost 750 kg, and the 75, 95, and 99% of exposure were about 250, 4000, and 11 000 kg, respectively (Fig. 1; Supplementary Table 1). Since pesticide use follows an extremely skewed distribution and the functional form of the relationship between pesticide use and birth outcomes is

unknown and could be non-linear, we report results using several measures of pesticide exposure.

We target our analyses on the effects of pesticide exposure to a subsample of regions that included 137 210 observations and had the highest level of pesticide exposure (Table 1, Supplementary Table 1), as well as the highest representation of mothers with less than a high school degree (37%) and mothers of Hispanic origin (68%). Possibly reflecting the "Latino Epidemiologic Paradox"[36], this subsample had the lowest average rate of adverse birth outcomes (Table 1). Below we refer to this subsample as the "focal sample".

**Main analysis**. In the focal sample, we find that the consequences of being in the high exposure group (95th percentile and above) vs. the low exposure group (all others) differed among the different birth outcomes (i.e., birth weight, gestation length, and birth abnormalities.) There was little effect of cumulative exposure over gestation or by EPA acute toxicity category on the probability of low birth weight (<2500 g, or ~5.5 lbs). We do find a statistically significant decrease in birth weight of about 0.4% or ~13 g with cumulative pesticide exposure and pesticide exposure in the first trimester for individuals in the high exposure group relative to the low exposure group (Table 2; Supplementary Table 2). For all results, statistical significance is based on two-tailed *t*-tests of the estimated regression coefficients where the standard errors are clustered at the zipcode level.

For gestational length and the probability of short gestation (<259 d or 37 weeks), there are negative effects of high pesticide exposure. Being in the high exposure group for cumulative pesticides over gestation reduces gestational length by ~0.1% or ~9 hours and increases the probability of preterm birth by ~8% (9.8% of births are preterm in the focal sample and thus a 0.00745 increase in the probability of preterm birth is equivalent to a 7.6% increase in this sample; Tables 1 and 2). High exposure to low acute toxicity chemicals over gestation reduces gestational length by ~0.1%, while exposure in the second trimester reduces the probability of preterm birth by ~5%.

For birth abnormalities, being in the high vs. low pesticide exposure group for cumulative pesticide use over gestation increased the probability of a birth abnormality by ~9% (5.8% of

**Table 1 Summary statistics by sample**

| | Full estimation sample | Focal sample |
|---|---|---|
| Birth weight | 3363 (534) | 3379 (529) |
| Gestation (d) | 276 (17) | 276 (16) |
| Total kg active ingredients (AI) | 748 (4111) | 2384 (8328) |
| Total kg AI of toxicity cat. 1 & 2 | 248 (3589) | 882 (7862) |
| Total kg AI of toxicity cat 3 & 4 | 333 (1235) | 981 (1730) |
| Mother's age <19 | 0.09 | 0.09 |
| Mother's age 19 – 24 | 0.34 | 0.33 |
| Mother's age 25 – 35 | 0.49 | 0.49 |
| Mother's age >35 | 0.08 | 0.08 |
| Mother's ed: <HS | 0.33 | 0.37 |
| Mother's ed: HS degree | 0.32 | 0.31 |
| Mother's ed: Some college | 0.22 | 0.22 |
| Mother's ed: 4 years college+ | 0.11 | 0.10 |
| Mother is non-Hispanic White | 0.29 | 0.26 |
| Mother is non-Hispanic Black | 0.04 | 0.02 |
| Mother is Hispanic | 0.58 | 0.68 |
| Mother is other race/ethnicity | 0.08 | 0.04 |
| Child is male | 0.51 | 0.51 |
| Mother smoked | 0.02 | 0.01 |
| Mother underweight (BMI<18.5)* | 0.03 | 0.03 |
| Mother normal weight (BMI 18.5 – 25)* | 0.39 | 0.38 |
| Mother overweight (BMI 25 – 30)* | 0.26 | 0.27 |
| Mother obese (BMI>30)* | 0.24 | 0.25 |
| Prenatal care: <5 visits | 0.03 | 0.03 |
| Prenatal care: 5 – 8 visits | 0.13 | 0.14 |
| Prenatal care: 9 – 14 visits | 0.59 | 0.62 |
| Prenatal care: 15 + visits | 0.21 | 0.19 |
| Low birth weight (<2500 g) | 0.049 | 0.045 |
| Preterm birth | 0.103 | 0.098 |
| Birth abnormalities | 0.064 | 0.058 |
| N | 692 586 | 137 210 |

Summary statistics for the "full estimation sample" of all births except those in 2006 (due to missing tobacco field) and the subsample of births born to mothers residing in regions with pesticides and births in each year ("focal sample"). Characteristics denoted with "*" indicate attributes that were only available from 2007 forward. Mean (standard deviation) are provided for birth weight, gestation and pesticide active ingredients, and rates within the population are provided for the remaining characteristics

births have a birth abnormality in this sample). Exposure by toxicity category and by trimester had no significant effect in the binary model.

Parsing pesticide use by aerial and ground application, we found being in the high exposure group for ground applications had a significant effect on log birth weight, log gestation, preterm birth, and birth abnormalities with magnitudes similar to those reported above (Table 3). Ground application represents roughly 95% of total active ingredients used, and thus high ground exposure represents most of the cumulative pesticide exposure measure (Supplementary Table 1).

**Robustness tests**. To explore if either inaccuracies in geocoding or spillover of pesticides from surrounding areas contaminated our results we excluded births for mothers living within 200 m of a PLS Section boundary. We found a similar overall pattern of statistical significance as in the larger sample. Although the magnitude of the coefficients increased, the effects on birth weight and gestational length remained <1%, and the effects on the probability of low birth weight, preterm birth, and abnormalities were at most 13% higher for the high exposure group relative to the low exposure group (Table 3).

We also estimated the trimester model including pesticide use in the "fourth trimester" (i.e., the 3 months following birth). As anticipated, exposure during the three months following birth did not have a significant effect on any outcomes observed at birth (Table 3). This "placebo" analysis indicates that our empirical results are unlikely to be caused by omitted trends or factors that are correlated with both pesticide applications and infant health.

To further ensure the robustness of our results and inference, we checked different exposure cutoffs (births in the top 75th, or top 99th percentiles pesticide exposure as the "high exposure" group) as well as a continuous measure of exposure (Supplementary Methods, Supplementary Table 3). The magnitude of effects was small and generally non-significant with the 75th percentile cutoff. Being in the top one percent of pesticide exposure led to an 11% increased probability of preterm birth, 20% increased probability of low birth weight, and ~30 g decrease in birth weight relative to lower exposure (<99th percentile).

We also evaluated models with different location fixed effects, different assumptions about clustering the standard errors to address spatial and temporal error correlation, different sample exclusion restrictions on gestational age and different calculations of trimester, as well as models with other environmental contaminants that can affect in utero infant health (ozone, carbon monoxide, and temperature; Supplementary Note 1, Supplementary Tables 4–7). Although the exact magnitude and patterns of significance did change with these different models, all models consistently reported similar effect sizes. Overall, we report over 100 coefficients in the main text, of which 19 are significant. It is noteworthy that in all these tests, only a single significant coefficient in one model has the opposite sign from that expected. The fact that only one of roughly 20 statistically significant coefficients has the wrong sign is consistent with the notion that our empirical estimates are not plagued (at least to a first-order) by omitted variable bias. Further, since we do not adjust $p$-values for multiple comparisons, the number of significant effects we report is an upper bound on the "true" number of significant effects. Applying a Bonferroni correction for multiple comparisons that accounts for five outcomes and up to five covariates of interest (for the trimester model with trimesters "0–4"), the $\alpha$-level for statistical significance would change from 0.05 to as small as 0.002 (0.05 divided by 25). The only three coefficients that remained statistically significant with this Bonferroni correction were those associated with a single covariate of interest, total pesticide exposure over the gestation ($\alpha_{corrected} = 0.01$). Of these, two were associated with preterm birth (Tables 2 and 3) and one with log gestation (Table 3).

## Discussion

Concerns about the effects of harmful environmental exposure on birth outcomes have existed for decades. Great advances have been made in understanding the effects of smoking and air pollution, among others, yet research on the effects of pesticides has remained inconclusive. While environmental contaminants generally share the ethical and legal problems of evaluating the health consequences of exposure in a controlled setting and the difficulties associated with rare outcomes, pesticides present an additional challenge. Unlike smoking, which is observable, or even air pollution, for which there exists a robust network of monitors, publicly available pesticide use data are lacking for most of the world. As a result, studies have typically been either highly correlative at coarse resolutions or have included a small number of subjects. Both constraints make it difficult to assess whether residential agricultural pesticide exposure has no effect

**Table 2 Effect of pesticide exposure on five birth outcomes**

|  | Log BW | Low BW | Log gestation | Preterm birth | Birth abnormalities |
|---|---|---|---|---|---|
| *A. Full Estimation Sample (all births excluding 2006)* | | | | | |
| Sum AI | −0.000613 | −4.02e-05 | −0.000522 | 0.00290 | 0.00200 |
|  | (0.00127) | (0.00157) | (0.000375) | (0.00184) | (0.00155) |
| Tox Cat 1 & 2 | −0.000466 | 0.000890 | 0.000143 | −0.000491 | −0.00248 |
|  | (0.00127) | (0.00149) | (0.000383) | (0.00188) | (0.00169) |
| Tox Cat 3 & 4 | 0.00126 | *−0.00319*** | −0.000447 | 0.00201 | −9.16e-05 |
|  | (0.00117) | *(0.00115)* | (0.000392) | (0.00180) | (0.00159) |
| Trimester 1 | −0.00144 | −0.000345 | −5.40e-05 | 0.000563 | 7.86e-05 |
|  | (0.00123) | (0.00121) | (0.000402) | (0.00200) | (0.00136) |
| Trimester 2 | −0.000485 | 0.000975 | −1.25e-05 | 0.00215 | −0.000157 |
|  | (0.00118) | (0.00130) | (0.000378) | (0.00163) | (0.00157) |
| Trimester 3 | 0.00225 | −0.00126 | −0.000123 | −0.00106 | −0.00170 |
|  | (0.00126) | (0.00152) | (0.000419) | (0.00196) | (0.00155) |
|  | $N = 692\,586$ | $N = 692\,586$ | $N = 692\,586$ | $N = 692\,586$ | $N = 688\,985$ |
|  | $R^2 = 0.06$ | $R^2 = 0.02$ | $R^2 = 0.04$ | $R^2 = 0.03$ | $R^2 = 0.04$ |
| *B. Focal sample (births in regions with births, pesticides in all years)* | | | | | |
| Sum AI | **−0.00387*** | 0.00213 | **−0.00132*** | **0.00745*** | **0.00504*** |
|  | **(0.00162)** | (0.00192) | **(0.000558)** | **(0.00275)** | **(0.00225)** |
| Tox Cat 1 & 2 | −0.00152 | 0.00224 | −0.000273 | 0.00318 | 0.00128 |
|  | (0.00169) | (0.00208) | (0.000504) | (0.00264) | (0.00193) |
| Tox Cat 3 & 4 | −0.000235 | −0.00275 | **−0.00135**** | 0.00432 | 0.00249 |
|  | (0.00182) | (0.00182) | **(0.000492)** | (0.00248) | (0.00236) |
| Trimester 1 | **−0.00362*** | 0.00146 | −0.000476 | 0.00276 | 0.00163 |
|  | **(0.00149)** | (0.00142) | (0.000548) | (0.00289) | (0.00175) |
| Trimester 2 | −0.00194 | **0.00363*** | −0.000647 | **0.00498*** | 0.00247 |
|  | (0.00132) | **(0.00147)** | (0.000442) | **(0.00221)** | (0.00189) |
| Trimester 3 | 0.000725 | −0.000161 | −0.000401 | 0.000780 | −0.000917 |
|  | (0.00153) | (0.00176) | (0.000532) | (0.00253) | (0.00204) |
|  | $N = 137\,210$ | $N = 137\,210$ | $N = 137\,210$ | $N = 137\,210$ | $N = 136\,621$ |
|  | $R^2 = 0.06$ | $R^2 = 0.03$ | $R^2 = 0.04$ | $R^2 = 0.03$ | $R^2 = 0.04$ |

Regression coefficients for three different measures of pesticide exposure in the full estimation sample (A) and the focal sample (B). All exposure variables are coded as binary variables with high exposure (>95% exposure based on all births) and low exposure (all others) groups. Sum AI indicates total kg of pesticide active ingredients applied in the 2.6 km$^2$ region encompassing mothers addresses, "Tox" indicates total kg of active ingredients by toxicity categories based on the EPA signal word, and trimester indicates total kg by trimester. All models include a set of covariates to control for characteristics of the mother and infant as well as regional and temporal dummy variables as described in the text (see Supplementary Table 2 for coefficients). These are omitted from the tables for visual clarity. For all tables, standard errors, clustered at mother's zipcode, are below the coefficients in parentheses. The statistical significance of two-tailed $t$-tests is indicated by * $p < 0.05$, ** $p < 0.01$, and *** indicates significance with a Bonferroni correction for multiple comparisons. Bold indicates significant coefficients in the expected direction and italics indicates the one significant coefficient with the opposite than expected sign. Values reported in the text are converted to percent or units of change (days, grams) based on rates of outcomes in the sample (Table 1). For example, 9.8% of births are preterm in the focal sample and thus a 0.00745 increase in the probability of preterm birth is equivalent to a 7.6% increase in this sample

or whether logistical and analytical barriers have obfuscated the identification of important effects.

Our study bridges the gap between detail and scale by leveraging vast pesticide and birth data for the San Joaquin Valley, CA. Our study has far stronger statistical power to identify effects than previous studies owing to over a hundred thousand birth observations, individual maternal and birth characteristics, and the inclusion of fine-scale regional and temporal fixed effects (PLS Township-year and birth month indicator variables). As a result of our statistical design, we have the analytical power to identify extremely small, but statistically significant negative effects of pesticide exposure on several birth outcomes, if they occur.

Furthermore, our study design and extensive pesticide data enable us to evaluate many details of the nature of pesticide exposure. For example, we can evaluate whether pesticide exposure in different trimesters or pesticides of different toxicity levels affected birth outcomes in different ways. Fetal susceptibility to environmental exposure varies through development[37]. Similarly, different chemical toxicity can have different expected health outcomes. Here we focused on aggregate chemicals grouped into high and low toxicity pesticides by their EPA Signal Word, which reflects acute toxicity. Acute toxicity does not necessarily indicate impacts from long-term exposure. As such, chemicals suspected to cause negative birth outcomes, such as organophosphates or atrazine would be classified as low toxicity. Nevertheless, we consistently find effects of less than a 10% increase in adverse outcomes for individuals in the top 5% of exposure regardless of timing or toxicity of exposure, even though which effects are statistically significant depends on the model.

Pesticide exposure has a highly skewed distribution in the San Joaquin Valley, where over half of births received no pesticides, the top quarter received about 250 kg and the top 5% received over 16 times that amount. Further, exposure to the top 25% levels had virtually no detectable effect whereas exposure to the top 1% had effects that were up to double the magnitude of effects observed for the top 5% of exposure. In other words, for most births, there is no statistically identifiable impact of pesticide exposure on birth outcome. Yet, for individuals in the top 5 percent of exposure, pesticide exposure led to 5–9% increases in adverse outcomes. The magnitude of effects were further enlarged for the top 1%, where these extreme exposures (>11 000 kg over gestation) led to an 11% increased probability of preterm birth, 20% increased probability of low birth weight, and ~30 g decrease in birth weight.

For perspective, other environmental conditions such as air pollution and extreme heat generally report a 5–10% increase in adverse birth outcomes, but from less extreme exposure[13, 15, 38]. Similar magnitudes of effects are also observed for other, non-exposure conditions of pregnancy. For example, stress during pregnancy may increase the probability of low birth weight by ~6%[39], while enrollment in supplemental nutrition programs is estimated to reduce the probability of low birth weight by a similar amount[40].

The significance of the negative effects of extreme pesticide exposure on birth outcomes is heightened by the fact that birth

**Table 3 Different model specifications evaluating effect of pesticide exposure on birth outcomes**

|  | Log BW | Low BW | Log gestation | Short gestation | Birth abnormalities |
|---|---|---|---|---|---|
| *A. Distance >200 m* | | | | | |
| Total AI | **−0.00611\*** | 0.00505 | **−0.00270\*\*\*** | **0.0128\*\*\*** | 0.00528 |
|  | **(0.00254)** | (0.00303) | **(0.000709)** | **(0.00362)** | (0.00288) |
| Tox Cat 1 & 2 | −0.00420 | 0.00321 | −0.00149 | 0.00589 | 0.00194 |
|  | (0.00263) | (0.00299) | (0.000791) | (0.00403) | (0.00287) |
| Tox Cat 3 & 4 | 0.00114 | −0.00367 | −0.000848 | 0.00588 | 0.00284 |
|  | (0.00266) | (0.00276) | (0.000751) | (0.00354) | (0.00298) |
| Trimester 1 | **−0.00633\*\*** | 0.00345 | −0.000960 | 0.00403 | 0.000888 |
|  | **(0.00213)** | (0.00229) | (0.000851) | (0.00426) | (0.00279) |
| Trimester 2 | −0.00210 | **0.00471\*** | −0.000461 | 0.00396 | 0.000110 |
|  | (0.00206) | **(0.00227)** | (0.000626) | (0.00354) | (0.00267) |
| Trimester 3 | 0.00131 | −0.00120 | **−0.00142\*** | 0.00413 | 0.00163 |
|  | (0.00213) | (0.00256) | **(0.000692)** | (0.00380) | (0.00257) |
|  | N = 74 646 | N = 74 646 | N = 74 646 | N = 74 646 | N = 74 338 |
| *B. Aerial & ground application* | | | | | |
| Total AI, Aerial | 0.000861 | 0.000945 | −0.000449 | 0.000808 | 0.00327 |
|  | (0.00183) | (0.00193) | (0.000761) | (0.00279) | (0.00265) |
| Total AI, Ground | **−0.00382\*** | 0.000917 | **−0.00132\*** | **0.00684\*** | **0.00485\*** |
|  | **(0.00157)** | (0.00180) | **(0.000571)** | **(0.00280)** | **(0.00211)** |
|  | N = 137 210 | N = 137 210 | N = 137 210 | N = 137 210 | N = 136 621 |
| *C. 0 & 4th trimester* | | | | | |
| Trimester 0 | 0.000308 | −0.000203 | 0.000158 | 0.00162 | 0.00106 |
|  | (0.00151) | (0.00205) | (0.000666) | (0.00323) | (0.00182) |
| Trimester 4 | −0.000666 | −0.00355 | 0.000348 | −0.00129 | −0.00228 |
|  | (0.00160) | (0.00188) | (0.000584) | (0.00282) | (0.00245) |
|  | N = 137 210 | N = 137 210 | N = 137 210 | N = 137 210 | N = 136 621 |

(A) Distance corresponds to mother's address being at least 200 m from the nearest region boundary, defined by the PLS Section. (B) "Aerial" and "Ground" are total pesticide active ingredients over gestation by application method. (C) is the same specification as in Table 2B with the addition of covariates for the first three months before pregnancy (trimester 0) and the first 3 months after birth (trimester 4). We expect the 4th trimester in particular to have no effect. * $p < 0.05$, ** $p < 0.01$, based on two-sided *t*-tests and *** indicates significance with a Bonferroni correction for multiple comparisons. Bold indicates significant coefficients in the expected direction

outcomes are persistent and costly. Reducing the incidence of adverse birth outcomes has obvious benefits for individuals, but also for society. Healthier babies require less intensive care as infants, have better long term health and are higher achieving in terms of earnings and employment[10, 41]. Thus, even small reductions in adverse outcomes can economically offset societal investment in programs such as supplemental nutrition programs offered to millions of low-income women[41].

Due to the concentration of negative outcomes at the very highest pesticide exposures, policies, and interventions that target the extreme right tail of the pesticide exposure distribution could largely eliminate the adverse birth outcomes associated with agricultural pesticide exposure documented in this study. As such, valuable and pressing future directions for research should focus on identifying the extreme pesticide users near human development and on the underlying causes for their extreme quantities of use. These insights are critical to designing appropriate and adaptive interventions for the population living nearby.

For instance, crops vary dramatically in their average pesticide use. Commodities such as grapes receive nearly 50 kg ha⁻¹ per year of insecticides alone in the San Joaquin Valley region[42], while other high value crops such as pistachios receive barely on third of that amount. Within these broad differences, there are also relevant differences among crops with regard to the chemical composition and seasonal timing of pesticide application. Finally, not all agricultural fields are in proximity to human settlement. Rather, as we illustrate, areas with consistent births and pesticides are a small fraction of the San Joaquin Valley. Thus, if extreme pesticide areas and vulnerable populations could be identified, strategies or interventions could be developed to mitigate the likelihood of extreme exposures.

One further difficulty is isolating the roles of individual chemicals and their mixtures in driving the negative outcomes. Doing so

is extremely challenging, because many chemicals are used in conjunction or in close spatial or temporal windows. Using a large-scale data-driven approach could provide a starting point from which individual or community based studies could be built. For example, statewide birth certificate data could enable the identification of potential hot spots of negative (and rare) birth outcomes while the Pesticide Use Reports provide a large sample of different pesticide mixtures. This could yield valuable information for targeting more detailed studies of individual exposures and difficult to observe outcomes (e.g., time to pregnancy, fetal deaths) towards regions and months of the highest concern.

There are some important limitations to our study. As with other environmental contaminants, controlled experiments evaluating the effects of pesticide exposure on birth outcomes are impossible due to clear ethical and legal constraints. This presents challenges both for interpretation and estimation. With regard to interpretation, we cannot observe all individual adaptive responses to pesticide use, such as staying indoors to avoid exposure to pesticide. Further, we can only observe the effects on live births. As a result, our estimates reflect both the direct effect of exposure on live births and the mitigating effects of avoidance behaviors. With regard to identification and estimation, establishing causality without random assignment into pesticide exposure relies on quasi-experimental approaches, such as the panel data models used here with observational data[43]. While there is no way to formally test if our methods have eliminated all sources of bias that preclude causal interpretation of the regression coefficients, our results are robust to multiple modeling approaches, including controlling for other environmental contaminants such as ambient concentration of pollutants and extreme temperatures. Similarly, we find no significant placebo effects of exposure in the 3 months following birth.

Birth records do not fully capture adverse outcomes such as abnormalities that are difficult to observe at birth nor are they comprehensive with regards to socio-demographics. Measurement error on the outcome variable would not bias our estimates of the effects of pesticide exposure unless it was somehow correlated with pesticide use, yet it could reduce our precision and thus the likelihood of finding statistical significance. With respect to sociodemographic factors, mothers that are exposed to extreme levels of pesticide are more likely to be minorities and have lower education than the sample population as a whole. While we control for these factors, there is potential for the high exposure sample to differ in other unobserved ways (e.g., unobserved maternal health behaviors) that could yield a higher likelihood of adverse birth outcomes. If so, this would result in overestimates of the effects of pesticide exposure on adverse birth outcomes.

Additionally, we measure pesticide exposure as all pesticide use (active ingredients) on production agriculture in the 2.6 km$^2$ PLS Section encompassing mothers' addresses. We do so because the diversity of chemicals applied in the San Joaquin Valley is extensive and the cumulative effects of multiple exposures are not well understood. However, some chemicals or combinations of chemicals may not be relevant to reproductive risk. Thus, our coefficients are likely underestimates for individuals exposed to a disproportionately high fraction of chemicals of reproductive concern for their PLS Section, year and birth month.

There is some indication that closer proximity to agricultural fields (<1000 m) results in increased odds of adverse birth outcomes[7, 20, 44]. For a study of this spatial and temporal breadth it is infeasible to directly measure distance from a sprayed field. However, for the San Joaquin Valley, PLS Sections that have any agriculture generally are agriculturally dominated[45]. Furthermore, the PLS Section is roughly 2276 m on a diagonal. Thus it is highly likely that the vast majority of households in PLS Sections with pesticide use are within 1000 m of a treated agricultural field. If pesticides dissipate much more rapidly, such that the effect is concentrated within 100 m of pesticide use, our study design would underestimate this relationship due to dilution with individuals living farther away from fields but still within the same PLS Section exposure. However, for this to be occurring, the population residing on-farm or adjacent to fields must be much smaller than the broader population residing in the San Joaquin Valley for us to observe such small coefficients. Indeed, this makes intuitive sense for California, where farmworkers overwhelmingly report living independent of their employers in houses or rental units, particularly if they have a spouse or children (93%)[46]. However, our results may under predict adverse birth outcomes in regions where a larger proportion of workers reside in employer-provided housing on or adjacent to fields, where a larger fraction of pesticide are applied aerially, or where permissible chemicals are more environmentally persistent or toxic to humans.

We also lack information on residence time at mother's address and employment. Much of the San Joaquin Valley economy is driven by the agricultural industry. If farmworkers were mostly migratory and followed the harvest, our measure of residential pesticide exposure would be inaccurate for this subset of the population. Yet, according to the National Agricultural Workers Surveys for 1996–2011, California farmworkers, especially if they have a spouse or children in their household, are settled (~98%)[46]. Our measure of exposure would also be artificially high if women were applying agricultural pesticides during pregnancy. While ~18% of California farmworkers are women, only 1.5% of women reported using pesticides in the past 12 months and 0% of women with a spouse or children had reported doing so[46]. Women could get additional exposure via their spouses, and ~16% of male farm workers reported loading, mixing or applying pesticides in the past year[46].

Finally, the San Joaquin Valley is well known to have substandard environmental quality, frequently exceeding EPA contamination levels for air quality. If such exposures co-vary with pesticide use and vary at small spatial and temporal scales, the coefficient on pesticide exposure could capture additional contamination despite our PLS Township-year and birth month controls. While we cannot be certain we have eliminated all sources of contamination that co-vary with pesticides, including a rich set of ambient air quality and temperature metrics did not change the basic results of this paper.

In conclusion, there is a growing literature on the relationship between pesticide exposure and adverse birth outcomes. Yet, evidence of a causal link between infant health and agricultural pesticide exposure remains uncertain due to small samples and lack of maternal or birth characteristics. Our study is the most comprehensive to date, bringing together the largest data file ever compiled on street-address level birth outcomes and fine scale exposure to agricultural pesticides. We provide robust evidence that there are multiple negative effects of residential agricultural pesticide exposure on adverse birth outcomes, but only for births exposed to very high levels of pesticides during gestation. The documented concentration of impacts in the extreme upper tail of the pesticide exposure distribution may explain why previous studies fail to consistently detect effects of pesticides on birth outcomes. Furthermore, the concentration of impacts in the extreme tail of the pesticide exposure distribution provides policy challenges and public health opportunities to balance these potentially serious but rare outcomes with the societal benefits of continued pesticide use.

## Methods

**Birth data**. Birth Data came from the California Birth Statistical Master File (BSMF) for 1997–2011. The BSMF includes individual birth records with information on characteristics of the mother (e.g., age, education, race), characteristics of the infant (e.g., sex, birth order), and birth outcomes for all births occurring in the state. While the BSMF existed prior to 1997, 1997 is the first year for which mother's street address is provided, a field critical to our refined measure of pesticide exposure. Further, 2011 is the last data year for which we had approval from California Committee for the Protection of Human Subjects and California Department of Public Health to evaluate the relationship between pesticides and adverse birth outcomes at the time of submission.

We geocoded singleton births to mothers residing in the San Joaquin Valley, CA (Fresno, Kern, Kings, Madera, Merced, San Joaquin, Stanislaus, and Tulare counties; 2) using the U.S. Census TIGER/Line shapefile for 2011 as the Address Locator in ArcGIS 10.0. Following Zhan et al.[47] we set spelling sensitivity to 50, minimum candidate score to 30, minimum match score to 70, and offset from the midline and end to 25 ft and 0 ft, respectively. Match rates for all years were 88–93%. Geocoded addresses were matched with the PLS Section (~2.6 km$^2$) encompassing the address for linking to the pesticide data. The projection for all files was North American Datum 1983 California Teale Albers.

We excluded births if gestation was <26 weeks or >50 weeks or if gestational age was missing ($n = 59\,226$). Gestational length for all observations was based on last menstrual period. From the sample with gestational age within the above limits, we excluded births if birth weight was reported as >5500 g, <250 g, or missing, or if total reported births (to the same mother) were >15 ($n = 746$). Since accurate assignment of location was of paramount importance to our measure of pesticide exposure, we further restricted the sample to match scores >85 after spot-checking matched addresses ($n = 24\,349$). Limiting our sample to singleton births to mothers residing in the San Joaquin Valley, CA with the above restrictions resulted in a sample of 753 290 births ("All births"). The binary cutoffs of pesticide exposure (i.e., 95%, 99%) were based on this sample. In 2006, the BSMF did not include data on maternal tobacco use and as a result, we dropped this entire year from the main analyses (though the inclusion of 2006 without tobacco as a covariate did not change our conclusions). The analysis sample, composed of all observations except those in 2006, was 692 589 ("full sample") and the analysis sample composed of observations that occurred in regions with pesticides and births in each year was 137 210 ("focal sample"; see Exposure, Econometric analysis, Supplementary Fig. 1).

**Pesticide data**. Pesticide data came from the California Department of Pesticide Regulation (DPR) Pesticide Use Reports (PUR) for 1996–2012. The PUR includes detailed temporal and geographical information on agricultural pesticide use,

including date of pesticide application, pounds of active ingredients (AI), method of application (ground, aerial), and EPA signal word (in decreasing toxicity: Danger or Danger-Poison (I), Warning (II), Caution (III), no word (IV)). The PUR data are available at the ~2.6 km$^2$ (1 mi$^2$) PLS Section, which is nested within ~93 km$^2$ (36 mi$^2$) PLS Townships.

The PUR data are internally checked for entry errors as well as outliers using multiple methodologies[48]. Outliers are flagged by the DPR if values exceed 50 times the median for a product–crop–year combination, or if use rates are >224 kg ha$^{-1}$ (200 lbs ac$^{-1}$) or 1121 kg ha$^{-1}$ (1000 lbs ac$^{-1}$) for fumigants. About 2% of records are found to have some type of error when checked by DPR, and entry errors are sent back to the county for correction while outliers that cannot be corrected by the county are adjusted to median use rate by product–crop–year[48, 49]. While the true error rate is unknown, the PUR data are widely considered highly accurate and the most accurate pesticide data in the world[49].

**Exposure**. Total pesticide application (kg of active ingredients) was summed over gestation and by trimester. Pesticide use by EPA signal word toxicity (grouped into high-I–II and low-III–IV) and by method of application (aerial, ground) was also summed over gestation. We aggregate chemicals grouped into high and low acute toxicity due to the large number of different chemicals applied in the San Joaquin Valley. To assess exposure we calculated gestation as the birth month and 8 previous months if the birth was on or after the 15th of the month and as the 9 preceding months if the birth was before the 15th of the month. We divided gestation into trimesters similarly, counting back from birth month. We counted backwards from birth month rather than forward from last menstrual period to reduce potential bias stemming from pesticide exposure altering the menstrual cycle[50] and thus trimester cutoffs based on last menstrual period.

We focused on binary measures of exposure, because the functional form of the relationship between pesticide exposure and birth outcomes is not well understood and could be nonlinear. We defined high exposure as being in the top 95th percentile or above based on all births with the above described sample restrictions, and low as all others. Thus, pesticide coefficients for such a binary comparison are interpreted as the effect of being in the high exposure group compared to the low exposure group. For robustness, we also estimated other model specifications using different binary cutoffs (99% as "high exposure" and 75% as "high exposure"). The 75th, 95th, and 99th percentile were based on the distribution of pesticide exposure for all births, and corresponded to 249, 4178, and 11 134, respectively (Supplementary Table 1). We also tried a continuous specification. The continuous, linear model makes the stronger assumption of a linear relationship between pesticides and birth outcomes, which would fail under the plausible scenario of exposure thresholds or saturation of effects. For completeness, we nevertheless tested a continuous specification. Finally, we explored a restricted cubic spline specification for cumulative pesticide exposure with knots at the 75, 95, and 99% of the pesticide distribution. Due to the extremely skewed pesticide distribution (Fig. 1), we continued with the binary split for computational and interpretational ease.

**Econometric analysis**. We created a panel dataset using the above individual birth data and PLS Section-level measures of pesticide exposure. The empirical analysis was based on the estimation of variants of the following regression model:

$$B_{itsyw} = \delta_0 + AI_{tsyw}\theta + X_{itsyw}\beta + \alpha_{ty} + \gamma_m + \varepsilon_{itsyw} \qquad (1)$$

where $i$ denotes the individual birth, $t$ denotes PLS Township, $s$ represents PLS Section, and $y$, $m$, and $w$ denote the year, month, and week of birth, respectively. $B$ is a birth outcome, such as log birth weight or an indicator of preterm birth. AI represents different binary measures of active ingredient use. Depending on the model, it can be a scalar equal to one if total pesticide AI use during gestation is in the top 5% of the sample and zero otherwise. Or it can be a vector of indicator variables for AI by trimester, or by toxicity category. The coefficient $\theta$ is thus the effect of being in the high exposure group compared to all other exposures. The vector $X$ represents individual level indicator variables for mother's age group (<19, 19–24, 25–35, 36+, missing), educational attainment (<high school, high school, some college, college or more, missing; Supplementary Note 1), race/ethnicity (non-Hispanic white, non-Hispanic Black, Hispanic, missing), tobacco use (no use, use, missing; Supplementary Note 1), number of prenatal care visits (<5, 5–8, 9–14, 15+, missing), infant's sex (male, female, missing), and whether it was first born. $\alpha_{ty}$ represents PLS Township-by-year indicator variables or "fixed-effects". These fixed-effects effectively de-mean attributes of a given birth by the PLS Township average in the birth year[43]. This accounts for time invariant characteristics shared by all births within a PLS Township in a given year that may be difficult to observe and could otherwise bias the estimation of $\theta$ (e.g., local economic activity and/or the quality of local health care providers). $\gamma_m$ represents a month fixed effect, used to control for seasonality differences in birth outcomes and pesticide exposure shared across locations and years. These fixed effects rely on variability in pesticide application within PLS Sections over time. Pesticide use is highly variable over time even at small spatial scales (Fig. 2), due in large part to on-farm decisions such as crop type and crop rotation, as well as other highly variable conditions such as weather. Finally, we used cluster robust standard errors, clustered at the zipcode, which allow for arbitrary serial correlation in the errors within zipcodes. For robustness, we evaluated models with zipcode-year rather

than PLS Township-year fixed effects and standard errors clustered at the PLS Township rather than zipcode.

We estimate this equation for five birth outcomes; log birth weight (g), log gestational age (d), low birth weight (<2500 g), preterm birth (<259 d), and birth abnormality (congenital anomaly and abnormal conditions/procedures; Supplementary Note 1, Supplementary Table 8). For very rare events, estimating linear probability models for binary outcomes can be problematic. However, none of these outcomes are especially rare in our sample (≥4.5%), and linear probability models are much more conducive to implementation of fixed effects models[43].

Controlling for individual level behavior and risk through the use of mother's fixed effects would have further reduced concerns about omitted variables bias. However, information that would have allowed us to identify births to the same mothers across different years in the California BSMF was unavailable. As a result, including mother's fixed effects was impossible. We controlled for a variety of observable characteristics in our estimation strategy, yet there are characteristics that separate agricultural, suburban, and urban dwellers in ways that are difficult to observe. To isolate a population with higher exposure (who may also be more affected by pesticide exposure) and more homogeneous observable characteristics, we constructed a subsample of births—births in PLS Sections with pesticides and births in all years ("focal sample").

For robustness, and to probe the underlying mechanisms of exposure, we ran additional models on the focal sample. First, we divided exposure into aerial and ground application and re-evaluated total exposure by application method for each of the five birth outcomes. Next, to check if either inaccuracies in geocoding or spillover of pesticides from surrounding areas contaminated our results, we reran the analysis with addresses in the interior of each PLS Section (at least 200 m from the PLS Section boundary), which excluded over 40% of each PLS Section area and associated births. We further tested models with gestation restricted to <44 weeks, removing the small fraction of pesticides that are categorized as adjuvants, and calculating trimester from date of last menses forward (Supplementary Table 7). We also analyzed models including pesticide exposure in the 3 months following birth (the "4th trimester") as a placebo test. Finally, we investigated the potential from confounding environmental pollution and temperature than can affect in utero infant health. It is well documented the San Joaquin Valley, CA has poor air quality[51]. Using data from the California Air Resources Board Ambient Air Quality Monitoring Stations as well as data from the National Climatic Data Center (NCDC) Global Historical Climatology Network-Daily (GHCN-Daily), we included binned covariates for quantiles of ambient ozone and carbon monoxide levels (<70, 70–90, 90–95, >99%) as well as temperature bins (45, 45–65, 65–85, >85 °F; 7.22, 7.22–18.33, 18.33–29.44, >29.44 °C) totaled over gestation and by trimester (Supplementary Note 1).

For ease of interpretation in the text, the coefficients are converted to percent change within the sample population. Thus, for binary outcomes (e.g., birth abnormalities), coefficients were scaled by the rate of the outcome in the focal sample and multiplied by 100. For example, a 0.005 increase in the probability of a positive outcome in the focal sample (Table 2), in which 5.8% of births have an abnormality (Table 1), is equivalent to an 8.6% increase in this outcome in this sample.

**Data availability**. Due to confidentiality requirements, the California Birth Statistical Master file analyzed in the current study is not publicly available. It can be obtained via application from the California Department of Public Health (https://archive.cdph.ca.gov/programs/ohir/Pages/OHIRApplications.aspx). All pesticide use data used in this manuscript are freely and publicly available to download from the California Department of Pesticide Regulation (http://www.cdpr.ca.gov/docs/pur/purmain.htm).

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

## Acknowledgements

We thank Kenneth Wachter and Peter Kareiva for insightful comments that improved the quality of this manuscript. We also thank the California Department of Pesticide Regulation and the California Department of Public Health for collecting and providing data. This research was conducted with CPHS (project #12-10-0865), UCSB IRB (project #15-0085), and UCB IRB (project #2015-09-7904) approvals. A.E.L. acknowledges funding from the UC President's Postdoctoral Fellowship.

## Author contributions

A.E.L. conceived and designed the study, designed and completed the analysis, and wrote the manuscript. S.D.G. conceived and designed the study, and wrote the manuscript. O.D. conceived and designed the study, designed the analysis, contributed environmental data products, and wrote the manuscript.

## Additional information

**Competing interests:** The authors declare no competing financial interests.

