## [Peer Review File · Nature Communications]

Reviewers' comments:

Reviewer #1 (Remarks to the Author):

This manuscript examines the relationship between agricultural pesticide exposure during pregnancy and adverse birth outcomes in the San Joaquin Valley of California. The authors found that those exposed to the highest levels of pesticides (top 1% and 5%) during the gestational period were associated with adverse outcomes including low birth weight, preterm birth and birth abnormalities.

This is the most comprehensive look at pesticides in this highly exposed area of California with regard to a wide array of adverse birth outcomes to date. This paper provides a broad look across a large study population with relatively detailed exposure assessment. The statistical analysis is sound and thorough and the introduction and discussion provide ample review of the previous literature.

The paper provides a broad look at pesticides in the area, but there remains to be some discussion of implications of these findings and the next steps. Do the authors have thoughts about looking into certain geographic hot spots or specific pesticides that may be driving these associations?

The manuscript is well-written and is of sufficient length. There are a few more methodological details that could be included and some minor clarifications:

- A list of "birth abnormalities" and distribution of what they encompass (e.g., which abnormal conditions/procedures from the birth certificate)
- Was gestational age calculated from LMP and/or obstetrical estimate (in later years)? Missing gestational age in weeks should be excluded if weight for age is not plausible.
- Why was gestational age included between 44-50 weeks? Also, how were the total reported births to the same mother >15 (N=746 excluded) obtained – is that across different years? On page 20, line 430-432, it is stated that information is not available.
- How was the focal sample selected? What geographic areas does it include? The methods mention regions, but what scale are those regions? Could that area be mapped?
- I am confused by the result of exposure in the 2nd trimester reducing preterm birth by 5% (page 8, line 160-162). Which table is this presented in?
- Clarification (page 8, line 158)... pesticides over gestation reduces gestational length by ~0.1% or ~9 h and increase the... Should that be "d" for days instead of h? If not, what does h stand for?
- Typo (page 20, line 427) LPMs→LMPs

Reviewer #2 (Remarks to the Author):

The datasets created and analysis provided by the authors explores an important area of population health concern. The approach taken is creative and allows a more detailed examination of the issues of pregnancy health and outcome.

The pesticide use reports include all types of agriculture chemicals; insecticides, fungicides, herbicides, etc. as well as other agents such as wetting agents and surfactants, which are mixed with the agents applied. How were these diverse agents managed in the descriptions of kg "pesticides" applied?

While the authors note that corrections for multiple comparisons were not done - it would be of interest to see the impacts of such an analysis.

Reviewer #3 (Remarks to the Author):

What are the major claims of the paper?

The topic of this paper is the investigation of the impact of agricultural pesticide use on reproductive outcomes **for the residents of the communities where these chemicals are used**. Authors correctly claim that theirs is one of the largest and most comprehensive investigations of this topic. They found that adverse reproductive outcomes were increased for residents of areas with the highest use of total pesticides.

Are they novel and will they be of interest to others in the community and the wider field?

Yes novel and yes of interest.

Novelty: While certainly not the first study of impact of agricultural pesticide use on nearby residents, this is the largest study—critical for assessment of relatively rare outcomes like reproductive anomalies visible at birth. Also, the study is likely to be estimating a lower rather than an upper bound effect... which is concerning in the sense that the impact on residents in high use areas might be larger than estimated. This is likely for several reasons: 1) the design investigates ALL pesticides combined—some of which may not be relevant individually to reproductive risk. Detecting an observable association in this setting is actually rather surprising and also concerning. 2) The outcome data base, while likely one of the best birth outcome file registries in the United States, is still subject to missing data, and incomplete reporting. These omissions probably diluted the estimated effects. This is because there is no reason to believe that such data problems vary systematically by pesticide exposure areas-- as the authors note. 3) The use of health status of the infant at birth, without further follow-up, does not fully capture the potential impact of toxic exposures on prenatal development. For example some heart / developmental anomalies/ failure to thrive are not obvious at birth and are therefore not being considered in this analysis. 4) Due to data limitations in reporting of fetal deaths, the report does not include gestations of less than 26 weeks and does not capture any effect on fetal deaths. Several other features of the report are novel and of great interest: 1) The stronger effects observed for ground vs. aerial application of pesticides is of considerable interest and logically supports the conclusion that there is local contamination that impacts residents. Aerial application would be more dispersed. This is a novel analytic contribution made possible by California's unique environmental data system. 2) The measured high levels of pesticide use (extremes of the exposure distribution) contributed to the positive study findings and provide new insight on where risks might be greatest and who might need protection. Thus studying the San Joaquin Valley which has geographic areas with very high exposure makes this study a good test of potential risk. In my opinion, the absence of effects at the lower "doses" does not weaken the study findings... as there is little data in the mid-range of exposure, and many considerations why smaller underlying effects might not have been detectable (see discussion at beginning of paragraph above).

Interest: a) Public Policy: There are currently active legislative debates (for example currently in Hawaii) where this evidence would be viewed as highly relevant to public policy and environmental regulation. b) Public Investment in Environmental Monitoring: This study illustrates the value of the large and enhanced data base on pesticide use and birth records maintained in California. The purpose of these data are ultimately to serve as a sentinel early warnings and to protect the public health, but also to address environmental justice issues that are related to place of residence. Other states/ regions are currently considering investment in this scale of bio/enviro- monitoring and this study is relevant to those investment decisions. c) Toxicology/ Experimental Science--- Some of the compounds that are in high use in California are already shown to have toxic effects in experimental studies. While this report does not address specific pesticides, it provides a model for studying them in a human population --- where a large data base makes it possible to observe effects that do not rise to catastrophic proportions. In fact, the same California data base has been used to zero in on the risk to residents in areas where the fungicide methyl bromide is used <http://dx.doi.org/10.1289/ehp.1205682> and also organophosphate and carbamate pesticide (Environmental Research Such chemical-specific results do not take away from the importance or novelty of the study reviewed here—but probably should be cited?

If the conclusions are not original, it would be helpful if you could provide relevant references.

As noted by the authors, most prior studies are weaker—as they note in their introduction where they cite a prior review of this topic.

Is the work convincing, and if not, what further evidence would be required to strengthen the conclusions?

--I think I answered this above and below--- I find the work convincing and valuable.

-- Next steps : New approaches to the study of mixture effects and more studies that address affects of mixtures are needed... as mixtures are the real way humans are exposed to pesticides. One strength of this study, as noted by the authors, is that this study measured total pesticide use. While there are those critics who might see this as a limitation, it can also be regarded as a strength. Studies that investigate one pesticide at time are subject to the criticism that other unmeasured pesticides or exposures are riding along with the specific pesticide investigated. (this is a very difficult challenge in this field!!)

On a more subjective note, do you feel that the paper will influence thinking in the field?

Environmental effects on health are highly politicized. Some will find these data convincing for the reasons I gave above.

Others might criticize this paper for what it cannot do as follows:

---cannot identify specific pesticides of concern (also discussed above under “Next Steps”- as this is also a strength!)

---cannot speak to mechanisms

---cannot speak to specific pregnancy problems/conditions that are most sensitive to exposures--
- this is not possible because vital status birth records are notoriously poor at reporting these specifically and reliably- so the author’s strategy to combine them is justifiable and in my opinion the best practice possible given the design and the data available.

---cannot report on the actual individual exposures and internal dose (blood, urine) of pesticides in relation to outcomes. As the authors note in their introduction, this would be costly and frankly unlikely to be done on the population scale addressed by this study. If the population is smaller it is not representative, if the population is large and representative (the San Joaquin Valley in this case), individual biomonitoring is likely unaffordable and not feasible.

---cannot completely eliminate alternative exposures that co-vary with residence location. This is an eternal limitation of human observational studies... but perhaps because this is an ecological analysis—might be a more apt criticism.

In other words: It is likely impossible to ever conduct a definitive study --- rather the question here, in my mind and likely will be shared by others, is whether this study was the best effort for the data available, and whether it enlightens the field. In my judgement it passes these tests.

Please feel free to raise any further questions and concerns about the paper.

There are some places where the writing might be softened. For example, the birth records used do not fully account for social factors that co-vary with exposures—the authors acknowledge that this as a limitation, but I would ask that they check for over-statement of their case that they have controlled for these factors see line 91 and line 114---- for example....actually the vital status data really do not provide a rich set of demographic characteristics—as per the supplementary data authors provide on covariables.. There is also the need to very directly discuss the limitations of ecological designs (of which this is one). I do not believe that the ecological design invalidates the study , however.

Table 1.... Please label what is tabled in the header (e.g. “Regression Coefficient (standard error)”-
- I don’t like having to

Lines 377-379.... Question about coding trimester: Please explain why this was the way you chose to calculate trimester of exposure—counting backwards from data of birth? Counting backwards from birth would misclassify long or short gestations? The boundaries of trimesters

really matter... especially between first and second... so this type of misclassification could obscure trimester effects? I might be missing something... but please clarify.

Question about 4th trimester placebo analysis: Seems that this was a risky "placebo" design if trimester 4 pesticide use is highly correlated with trimester 1-3 pesticide use? I might have missed this... but did you report on this? Just curious—was there a correlation between trimester and pesticide use.... That could suggest a relationship of pesticide use with fetal loss or longer time to pregnancy? Just wondering... Wouldn't expect a correlation unless seasons of high use resulted in fewer pregnancies that lasted > 26 weeks...?

The very low rates of maternal smoking (Table 2) may relate to reluctance to admit prenatal smoking given public stigma --- but the cultural norms of the San Joaquin Valley Hispanic population might also explain this? Can the author's comment on this?

We would also be grateful if you could comment on the appropriateness and validity of any statistical analysis, as well the ability of a researcher to reproduce the work, given the level of detail provided.

I think that the supplementary data plus the methods section would make it possible for someone to replicate the findings. Details on coding are presented as are alternative model specifications.

I found the analytical choices made in this paper appropriate to the strengths and limitations of their data. This is rather refreshing... as sometimes analysts fail to see the flaws in insisting on classic dose response when the data just are not suited to such analysis. This is evident and available to the reader in Figure 1 and Supplementary Table 1 where the percentiles of pesticide exposure clearly show highly skewed distributions.

The authors did an excellent job of evaluating the most logical alternative explanations for their findings that are available to them—at the individual level, maternal smoking, education, prenatal care visits, at the community level, air pollution known to be a severe problem in the San Joaquin Valley.

Point-by-Point Response to Reviewers

We sincerely appreciate the depth and breadth of the comments by the three reviewers and the opportunity to revise our manuscript accordingly. Addressing these comments has substantially improved the quality and clarity of the paper. Further, they have sharpened our thinking about the project and productive future directions for our work and the field more broadly. The reviewers' comments are below in italics with specific comments highlighted in bold and point-by-point responses in regular text.

Responses to Reviewer 1 Comments on:

“Agricultural Pesticide Use and Adverse Birth Outcomes in the San Joaquin Valley, CA”

Reviewer 1 comments:

This manuscript examines the relationship between agricultural pesticide exposure during pregnancy and adverse birth outcomes in the San Joaquin Valley of California. The authors found that those exposed to the highest levels of pesticides (top 1% and 5%) during the gestational period were associated with adverse outcomes including low birth weight, preterm birth and birth abnormalities.

This is the most comprehensive look at pesticides in this highly exposed area of California with regard to a wide array of adverse birth outcomes to date. This paper provides a broad look across a large study population with relatively detailed exposure assessment. The statistical analysis is sound and thorough and the introduction and discussion provide ample review of the previous literature.

*The paper provides a broad look at pesticides in the area, but there remains to be some discussion of implications of these findings and the next steps. **Do the authors have thoughts about looking into certain geographic hot spots or specific pesticides that may be driving these associations?***

Thank you for this comment. One of the primary challenges in this field is addressing exposures to multiple chemicals that occur within small spatial or temporal windows. Here we pooled all active ingredients (or parsed by EPA toxicity categories) because the interactions between chemicals are poorly understood. However, the reviewer is correct that to isolate mechanisms, we need to better understand which chemicals or mixtures of chemicals are driving the observed relationships. We have added a paragraph in the discussion (lines 271-295) to stimulate future research in this regard.

The discussion of future directions now reads:

“Due to the concentration of negative outcomes at the very highest pesticide exposures, policies and interventions that target the extreme right tail of the pesticide exposure distribution could largely eliminate the adverse birth outcomes associated with agricultural pesticide exposure documented in this study. As such, valuable and pressing future directions for research should focus on identifying the extreme pesticide users near human development and on the

underlying causes for their extreme quantities of use. These insights are critical to designing appropriate and adaptive interventions for the population living nearby.

For instance, crops vary dramatically in their average pesticide use. Commodities such as grapes receive nearly 50 kg ha⁻¹year⁻¹ of insecticides alone in the San Joaquin Valley region (Larsen & Noack *in press*), while other high value crops such as pistachios receive barely 1/3 of that amount. Within these broad differences, there are also relevant differences between crops with regard to the chemical composition and seasonal timing of pesticide application. Lastly, not all agricultural fields are in proximity to human settlement. Rather, as we illustrate, areas with consistent births and pesticides are a small fraction of the San Joaquin Valley. Thus, if extreme pesticide areas and vulnerable populations could be identified, strategies or interventions could be developed to mitigate the likelihood of extreme exposures.

One further difficulty is isolating the roles of individual chemicals and their mixtures in driving the negative outcomes. Doing so is extremely challenging, because many chemicals are used in conjunction or in close spatial or temporal windows. Using a large-scale data-driven approach could provide a starting point from which individual or community based studies could be built. For example, statewide birth certificate data could enable the identification of potential hot spots of negative (and rare) birth outcomes while the Pesticide Use Reports provide a large sample of different pesticide mixtures. This could yield valuable information for targeting more detailed studies of individual exposures and difficult to observe outcomes (e.g. time to pregnancy, fetal deaths) towards regions and months of the highest concern.”

The manuscript is well-written and is of sufficient length. There are a few more methodological details that could be included and some minor clarifications:

• A list of “birth abnormalities” and distribution of what they encompass (e.g., which abnormal conditions/procedures from the birth certificate)

Thank you for this comment. We have added a supplementary table (Table S8) with the rates of different abnormal conditions and procedures listed in the Birth Statistical Master File (also attached at the end of this document). Some of the specific conditions and classification procedures used by the California Department of Public Health changed in 2006, which is one motivation for creating a binary outcome for abnormal procedures/conditions as we describe in the SI text. Where that occurred, we reported rates (Table S8) based on the 2006-2011 population.

• Was gestational age calculated from LMP and/or obstetrical estimate (in later years)?

For all observations, gestational age was calculated from LMP. We have clarified this in lines 394-395.

• Missing gestational age in weeks should be excluded if weight for age is not plausible.

Any observation missing gestational age was dropped from the analysis, as was any observation missing birth weight. We describe these and other sample restrictions in the methods section (lines 395-401).

• Why was gestational age included between 44-50 weeks?

Thank you for this comment. We agree that including births with gestational age between 44-50 weeks and excluding births with gestational age beyond 50 weeks is somewhat arbitrary. A first point is that births with gestational age longer than 43 weeks are rare: they account for less than 4% of births in the San Joaquin region. The justification for including births with gestational age up to 50 weeks is that pesticide exposure may cause changes in the menstrual cycle (e.g. Farr et al. 2004) and our constructed measure of gestational age depends on the reported date of last menses. Thus, we did not want to exclude the possibility that gestation reported for > 43 weeks from LMP was a valid observation for this population.

To evaluate the importance of the gestational age exclusion criteria, we re-estimated our main analysis excluding gestational age above 43 weeks as suggested by the reviewer. The patterns of significance and the magnitude of the effects were qualitatively similar regardless of the gestational age exclusion criteria. We have included this table in the supplementary material, Table S7 panel B (also copied at the end of this document). Births from gestational periods longer than 50 weeks are extremely rare (less than 1% of total births) and clearly their inclusion or exclusion from the analysis is not consequential.

S.L. Farr, G.S. Cooper, J. Cai, D.A. Savitz, D.P. Sandler. 2004. Pesticide use and menstrual cycle characteristics among premenopausal women in the Agricultural Health Study. *Am J Epidemiol* 160, 1194-1204.

• Also, how were the total reported births to the same mother >15 (N=746 excluded) obtained – is that across different years? On page 20, line 430-432, it is stated that information is not available.

The number on previous births born to the mother is available as a field in the Birth Statistical Master File. However, there is no information (e.g. driver's license number, or SSN) that would enable us link siblings.

• How was the focal sample selected? What geographic areas does it include? The methods mention regions, but what scale are those regions? Could that area be mapped?

Thank you for pointing out the lack of clarity in our description. We have addressed this comment in several ways. First, we added an inlay of the state of California to Figure 2 to illustrate the scale of the study region and its location within the state. Also illustrated in Figure 2 are the Public Land Survey Townships (93km²) and Sections (2.6km²). The finest resolution of the pesticide data is at the Section level.

Figure 2 with the added inlay illustrating the location of the San Joaquin Valley within California. Illustrated in the inlay on the left, is pesticide use at the finest available scale (2.6km²; Public Land Survey Section). Exposures were based on monthly totals of pesticide use at the Section scale. The main figure illustrates total active ingredients and variation in active ingredients at the Township scale (93 km²) for visual ease. Township-year controls (dummy variables or “fixed effects”) were included in all models to account for unobservable characteristics unique to births within a Township in a given year.

Second, as suggested by the reviewer, we have mapped the different geographical areas underlying our different samples (Figure S1). The full estimation sample was the universe of births between 1997-2011 to California residents that were successfully geocoded, had non-missing gestation, birth weight and covariates (restrictions described in lines 395-408, illustrated in gray in Figure S1). We then created a subsample of births in regions of consistent pesticide use (“focal sample”). As we describe in lines 482-490, this was to create a sample with more homogeneous characteristics. We illustrate this sample in black in Figure S1.

Figure S1. The distribution of PLS Sections included in the full estimation sample (gray) and the focal sample (black).

• *I am confused by the result of exposure in the 2nd trimester reducing preterm birth by 5% (page 8, line 160-162). Which table is this presented in?*

We have clarified this in the text (lines 164, 167, 170). The estimate of a 5% effect size is based on Table 1, panel B. Note that 9.8% of births are preterm in the focal sample (see Table S1) so the percent can be calculated as $(\text{coefficient}/9.8) * 100$. Here it is $(0.00498/9.8)*100 \approx 5\%$.

• *Clarification (page 8, line 158)... pesticides over gestation reduces gestational length by ~0.1% or ~9 h and increase the... Should that be “d” for days instead of h? If not, what does h stand for?*

We apologize for the lack of clarity, “h” represented hours. This has been replaced in the main text.

• *Typo (page 20, line 427) LPMs→LMPs*

We recognize the possible confusion between LMP and LPM (linear probability model) in this paper. We have removed LPM as an acronym.

**Responses to Reviewer 2 Comments on:
“Agricultural Pesticide Use and Adverse Birth Outcomes in the San Joaquin Valley, CA”**

Reviewer 2 comments:

The datasets created and analysis provided by the authors explores an important area of population health concern. The approach taken is creative and allows a more detailed examination of the issues of pregnancy health and outcome.

The pesticide use reports include all types of agriculture chemicals; insecticides, fungicides, herbicides, etc. as well as other agents such as wetting agents and surfactants, which are mixed with the agents applied. How were these diverse agents managed in the descriptions of kg "pesticides" applied?

Thank you for this insightful comment. The California Pesticide Use Reports (PUR) include all pesticides used on agriculture. Here we aggregate all active ingredients of pesticides used on production agriculture. We use “kg of active ingredients” rather than “kg of product” to focus solely on pesticide chemicals. Prior to 2002 adjuvants (e.g. surfactants, wetting agents) were listed as active ingredients and afterwards, they were not. Adjuvants account for less than 1% of pesticides over gestation. We recalculated to exposure excluding adjuvants, and re-ran our main models. Excluding adjuvants did not change our results (Table S7 Panel C, copied at the end of this document). We added total active ingredient (and AI by trimester) excluding adjuvants to the summary statistics in Table S1, along with an explanation in the table notes.

While the authors note that corrections for multiple comparisons were not done - it would be of interest to see the impacts of such an analysis.

In our analysis we have five birth outcomes and between one and five covariates of interest, depending on the model (1 for total pesticide exposure, 3 for trimester, 5 for trimester with trimesters “0” & “4” included). Thus, the bar for statistical significance as judged by a Bonferroni correction is extremely high and likely to result in false negatives. Nevertheless, we recalculated statistical significance to account for multiple comparisons as requested by the reviewer. We now note both the challenge of correcting for multiple comparisons and the result of doing so in the text. Lines 207-215 read, “Further, since we do not adjust p-values for multiple comparisons, the number of significant effects we report is an upper bound on the “true” number of significant effects. Applying a Bonferroni correction for multiple comparisons that accounts for five outcomes and up to five covariates of interest (for the trimester model with trimesters “0”-“4”), the corrected α -level would be as small as 0.002 ($\alpha_{corrected} = \frac{\alpha}{\#Cov.*\#outcomes} = \frac{0.05}{25}$). The only three coefficients that remained statistically significant with such a correction were those associated with total pesticide exposure over the gestation period ($\alpha_{corrected} = 0.01$). Of these, two were associated with preterm birth (Table 1, Table 3) and one with log gestation (Table 3).”

**Responses to Reviewer 3 Comments on:
“Agricultural Pesticide Use and Adverse Birth Outcomes in the San Joaquin Valley, CA”**

Reviewer 3 comments (Abridged):

The topic of this paper is the investigation of the impact of agricultural pesticide use on reproductive outcomes for the residents of the communities where these chemicals are used. Authors correctly claim that theirs is one of the largest and most comprehensive investigations of this topic. They found that adverse reproductive outcomes were increased for residents of areas with the highest use of total pesticides.

*Novelty: While certainly not the first study of impact of agricultural pesticide use on nearby residents, this is the largest study—critical for assessment of relatively rare outcomes like reproductive anomalies visible at birth. **Also, the study is likely to be estimating a lower rather than an upper bound effect... which is concerning in the sense that the impact on residents in high use areas might be larger than estimated. This is likely for several reasons.***

Thank you for this detailed comment. The reviewer kindly elaborated on several different ways our estimates may be underestimates, including data limitations in the Birth Certificate (e.g. under recording birth abnormalities) and pesticide data (e.g. some chemicals in pesticides are not relevant to reproductive risk). We have expanded our discussion of our results regarding the ways the estimates could be a lower bound. We acknowledge the data limitations of birth records (lines 310-311), as well as the fact that not all pesticides are relevant to reproductive risk (lines 321-327). We include an explanation that that could result in underestimates for individuals with above average exposure to chemicals of reproductive concern (lines 321-327). We also clarify that we can only observe the effects on live births (lines 299-301) and include discussion of future directions related to difficult to observe outcomes in lines 287-295.

Several other features of the report are novel and of great interest: 1) The stronger effects observed for ground vs. aerial application of pesticides is of considerable interest and logically supports the conclusion that there is local contamination that impacts residents. Aerial application would be more dispersed. This is a novel analytic contribution made possible by California’s unique environmental data system. 2) The measured high levels of pesticide use (extremes of the exposure distribution) contributed to the positive study findings and provide new insight on where risks might be greatest and who might need protection. Thus studying the San Joaquin Valley which has geographic areas with very high exposure makes this study a good test of potential risk. In my opinion, the absence of effects at the lower “doses” does not weaken the study findings... as there is little data in the mid-range of exposure, and many considerations why smaller underlying effects might not have been detectable (see discussion at beginning of paragraph above).

Thank you for this kind comment.

In fact, the same California data base has been used to zero in on the risk to residents in areas where the fungicide methyl bromide is used <http://dx.doi.org/10.1289/ehp.1205682> and also organophosphate and carbamate pesticide (Environmental Research Such chemical-specific

results do not take away from the importance or novelty of the study reviewed here—but probably should be cited?

We have added citations to studies that focus on specific chemicals (methyl bromide and organophosphate) in the same study regions. These citations are now included in the introduction (# 32, 35). Both of found exposure had ambiguous effects on birth outcomes. For example, Eskenazi et al. 2004 reported a decrease in gestational duration, but an unexpected increase in body length and head circumference, while Gemmill et al. 2013 reported a decrease in birth weight and head circumference, but an increase in gestational age. Parsing exposure to individual chemicals is extremely difficult, as noted by the reviewer.

Gemmill, A., Gunier, R.B., Bradman, A., Eskenazi, B. & Harley, G. (2013). Residential proximity to methyl bromide use and birth outcomes in an agricultural population in California. *Environ Health Perspect* 121: 737-743.

Eskenazi, B. Harley, K., Bradman, A., Weltzien, E., Jewell, N.P., et al. (2004). Associations of in utero organophosphate pesticide exposure and fetal growth and length of gestation in an agricultural population. *Environ Health Perspect* 112: 1116-1124.

There are some places where the writing might be softened. For example, the birth records used do not fully account for social factors that co-vary with exposures—the authors acknowledge that this as a limitation, but I would ask that they check for over-statement of their case that they have controlled for these factors see line 91 and line 114---- for example...actually the vital status data really do not provide a rich set of demographic characteristics—as per the supplementary data authors provide on covariables.

We have softened the language regarding birth records. We were trying to emphasize the availability of individual characteristics, but as the reviewer notes, data available on the birth certificate are not exhaustive.

There is also the need to very directly discuss the limitations of ecological designs (of which this is one). I do not believe that the ecological design invalidates the study, however.

Thank you for this comment. We have greatly expanded our discussion of the limitations of our study design. While we have individual level outcome variables and several individual level controls, exposure would be the same for two individuals from the same Section born in the same half of the same month. As the reviewer correctly noted, this could lead to underestimates for individuals that receive disproportionate levels of pesticides within their Section or disproportionate levels of pesticides of reproductive concerns. We have highlighted this in lines (321-327, 334-336). We also elaborate on other limitations of our study in the discussion.

Table 1.... Please label what is tabled in the header (e.g. “Regression Coefficient (standard error)”-- I don’t like having to read the table notes to find out what is tabled—might confuse other readers too.

Thank you for this comment. We have clarified what is tabled in each of the main tables.

Lines 377-379.... Question about coding trimester: Please explain why this was the way you chose to calculate trimester of exposure—counting backwards from data of birth? Counting backwards from birth would misclassify long or short gestations? The boundaries of trimesters really matter... especially between first and second... so this type of misclassification could obscure trimester effects? I might be missing something... but please clarify.

Thank you for this comment. As we described in the methods, we wanted to minimize the potential endogeneity between gestational age and pesticides. In other words, if pesticides lead to shorter gestation, the measure of pesticide exposure just during gestation would be influenced by this effect. We chose to work from birth month backwards to further avoid the potential that pesticides, which include endocrine disruptors, influence patterns of menstrual cycles (e.g. length of cycle, missed periods, etc; e.g. Farr et al. 2004) and thus further blur estimates of trimester cutoffs based on last menstrual cycle. We add lines 430-432 describing this.

Nevertheless, to further address the reviewer's comment, we re-visited our analysis calculating trimester and total exposure from date of last menses forward. We include this analysis as part of panel D in the supplementary Table S7. The patterns are qualitatively similar as with the calculation of date of birth backwards. For birth weight, the second trimester exposure was also found to be significantly positive, though the magnitude was not statistically different than that of the original analysis. This analysis, and the other new analyses (Table S7 panels B,C), consisted of another 30 model runs and 75 coefficient estimates (on covariates of interest) of which 20 were found to be statistically significant ($p < 0.05$). A single significant coefficient in one model had the opposite sign from that expected. The fact that only one of 20 statistically significant coefficients in this supplementary analysis has the wrong sign is consistent with the notion that our empirical estimates are not plagued by omitted variable bias.

S.L. Farr, G.S. Cooper, J. Cai, D.A. Savitz, D.P. Sandler. 2004. Pesticide use and menstrual cycle characteristics among premenopausal women in the Agricultural Health Study. *American Journal of Epidemiology* 160, 1194-1204.

Question about 4th trimester placebo analysis: Seems that this was a risky “placebo” design if trimester 4 pesticide use is highly correlated with trimester 1-3 pesticide use? I might have missed this... but did you report on this? Just curious—was there a correlation between trimester and pesticide use.... That could suggest a relationship of pesticide use with fetal loss or longer time to pregnancy? Just wondering... Wouldn't expect a correlation unless seasons of high use resulted in fewer pregnancies that lasted > 26 weeks...?

Thank you for this interesting comment. We have tried to address it in a couple of ways. First, we acknowledge that the amount of pesticide exposure is correlated with month of birth due to the seasonal patterns in the agricultural cycle. We illustrate this in the summary statistics (Table S2) and control for it using month dummy variables. We have added further explanation to the methods regarding the relationship between pesticide exposure and month of birth.

Second, we try to address the reviewer's curiosity regarding season of birth, exposure and number of births. Below is a figure representing pesticide exposure by trimester for each birth

month. On the x-axis is month of birth (January =1, February =2, etc.) and on the y-axis is kg of active ingredients. The shading represents the component of total exposure from the different trimesters. Above the bars is the aggregate number of births (in thousands) for each birth month.

The patterns of pesticide use by trimester fit the patterns of agricultural production in the region. For example, November, December, and January births have little 3rd trimester exposure, but have high first trimester exposure (since this corresponds to late spring, early summer) when pre-planting soil fumigation and early plant growth occurs.

There are a couple of things to note. First, overall pesticide use is correlated with birth month as we discussed earlier. Second, pesticide use by trimester is not highly correlated across trimesters. In other words, birth months that have high first trimester pesticide exposure do not necessarily have high second trimester exposure. Third, it does not appear that months with high pesticide exposure have a lower number of births. For example, the months with the most births (August, September, and July) are also those that receive the highest total exposure over the gestational period.

The very low rates of maternal smoking (Table 2) may relate to reluctance to admit prenatal smoking given public stigma --- but the cultural norms of the San Joaquin Valley Hispanic population might also explain this? Can the author’s comment on this?

As the reviewer notes, reported smoking rates are lower than commonly reported in other regions. According to the California Department of Public Health Maternal and Infant Health Assessment Survey, an annual population survey of California resident women, modeled after the CDC PRAM survey, 2.5% of women reported smoking during pregnancy in 2012-2013 (see p.3 of below link). This varied by race with 4.4% of white women and 1.3% of Hispanic reporting having smoked during pregnancy. The majority of women having children in the San Joaquin Valley are Hispanic, and we report 1-2% smoke rates, depending the sample.

Of course, both of these datasets could be biased by reluctance to admit prenatal smoking. When we removed smoking as a covariate (in order to include 2006 data that did not have smoking as a field), it did not change our main results. Thus, even if smoking is under reported, it is unlikely to be problematic for our conclusions.

<http://www.cdph.ca.gov/data/surveys/MIHA/MIHAPublications/MaternalSmokingFactSheet.pdf>

I think that the supplementary data plus the methods section would make it possible for someone to replicate the findings. Details on coding are presented as are alternative model specifications. I found the analytical choices made in this paper appropriate to the strengths and limitations of their data. This is rather refreshing... as sometimes analysts fail to see the flaws in insisting on classic dose response when the data just are not suited to such analysis. This is evident and available to the reader in Figure 1 and Supplementary Table 1 where the percentiles of pesticide exposure clearly show highly skewed distributions.

The authors did an excellent job of evaluating the most logical alternative explanations for their findings that are available to them—at the individual level, maternal smoking, education, prenatal care visits, at the community level, air pollution known to be a severe problem in the San Joaquin Valley.

Thank you for your kind and insightful comments.

Other Changes

To improve interpretation of the histogram figure, we changed the bin width to 5,000 kg rather than illustrate a set number of bins in the figure. Further, we weighted the pesticide distribution by the frequency of annual births. These changes had minor effects of the histogram (Figure 1, copied below).

Table S7. Model specifications evaluating the effect of pesticide exposure on birth outcomes— A) Original, B) with gestation <44wks, C) removing adjuvants, D) Gestation less than 44 weeks, no adjuvants, calculating trimester (and thus total) from menses forward. For all results, the coefficient is above the standard errors (in parentheses).

	Log BW	Low BW	Log gestation	Short Gestation	Any Abnormalities
A. Original					
Sum AI	-0.00387* (0.00162)	0.00213 (0.00192)	-0.00132* (0.000558)	0.00745** (0.00275)	0.00504* (0.00225)
Trimester 1	-0.00362* (0.00149)	0.00146 (0.00142)	-0.000476 (0.000548)	0.00276 (0.00289)	0.00163 (0.00175)
Trimester 2	-0.00194 (0.00132)	0.00363* (0.00147)	-0.000647 (0.000442)	0.00498* (0.00221)	0.00247 (0.00189)
Trimester 3	0.000725 (0.00153)	-0.000161 (0.00176)	-0.000401 (0.000532)	0.000780 (0.00253)	-0.000917 (0.00204)
N	137210	137210	137210	137210	136621
B. Gestation < 44wks					
Sum AI	-0.00443* (0.00171)	0.00252 (0.00202)	-0.00145** (0.00055)	0.00767** (0.00283)	0.00516* (0.00239)
Trimester 1	-0.00394* (0.00153)	0.00180 (0.00147)	-0.00078 (0.00049)	0.00300 (0.00295)	0.00165 (0.00181)
Trimester 2	-0.00219 (0.00137)	0.00377* (0.00149)	-0.00077 (0.00043)	0.00517* (0.00227)	0.00271 (0.00194)
Trimester 3	0.000424 (0.00155)	-0.00009 (0.00182)	-0.00054 (0.00048)	0.00085 (0.00257)	-0.00108 (0.00210)
N	133,690	133,690	133,690	133,690	133,118
C. No Adjuvants					
Total AI	-0.00406* (0.00156)	0.00213 (0.00190)	-0.00137* (0.00056)	0.00720** (0.00274)	0.00551* (0.00230)
Trimester 1	-0.00348* (0.00155)	0.00119 (0.00147)	-0.00042 (0.00057)	0.00247 (0.00297)	0.00156 (0.00178)
Trimester 2	-0.00196 (0.00132)	0.00351* (0.00153)	-0.00072 (0.00044)	0.00513* (0.00220)	0.00279 (0.00195)
Trimester 3	0.00070 (0.00155)	-0.00011 (0.00176)	-0.00031 (0.00054)	0.00020 (0.00252)	-0.00121 (0.00208)
N	137210	137210	137210	137210	136621
D. Gestation<44wks, No adjuvants, Gestation determined from menses,					
Total AI	-0.00226 (0.00171)	0.00163 (0.00222)	-0.00132** (0.000501)	0.00756** (0.00243)	0.00488* (0.00225)
Trimester 1	-0.00332* (0.00137)	0.00288 (0.00160)	-0.00065 (0.00050)	0.00401 (0.00265)	0.00262 (0.00187)
Trimester 2	-0.00307* (0.00121)	0.00382* (0.00155)	-0.00096 (0.00049)	0.00395 (0.00260)	0.00353* (0.00176)
Trimester 3	0.00263 (0.00171)	-0.00365* (0.00170)	0.00066 (0.00044)	-0.00184 (0.00254)	-0.00348 (0.00214)
N	133,632	133,632	133,632	133,632	133061

Table S8. List of birth abnormalities and their rate per 100,000 births.

Years	Code	Description	Rate per 100k births
97-11	73	NICU admission	3681.6
97-11	75	Other conditions/procedures not listed	2316.4
97-11	74	Newborn transferred to another facility within 24 hours of delivery	727.1
97-11	71	Assisted ventilation required immediately following delivery	433.6
06-11	87	Antibiotics received by the newborn for suspected neonatal sepsis	274.6
97-11	66	Significant birth injury (skeletal fractures, peripheral nerve injury, soft tissue or solid organ hemorrhage which requires intervention)	130.9
06-11	85	Assisted ventilation required for more than 6 hours	94.5
97-11	62	Additional and unspecified congenital anomalies (not listed)	35.8
97-11	30	Cleft palate with cleft lip	22.8
06-11	79	Gastroschisis	22.3
97-11	57	Down's syndrome (karyotype confirmed)	17
97-11	29	Cleft lip alone	15.1
06-11	86	Newborn given surfactant replacement therapy	14.5
06-11	81	Down's syndrome (karyotype pending)	13.3
97-11	28	Cleft palate alone	8.9
97-11	35	Hypospadias	8.6
06-11	83	Suspected chromosomal disorder (karyotype pending)	7.2
97-11	1	Anencephaly	6.8
97-11	2	Meningiomyelocele/Spina bifida	6.8
06-11	76	Cyanotic congenital heart disease	6.6
97-11	70	Seizure or serious neurological dysfunction	6.2
06-11	80	Limb reduction defect (excluding congenital amputation and dwarfing syndromes)	5.7
06-11	77	Congenital diaphragmatic hernia	3.3
06-11	82	Suspected chromosomal disorder (karyotype confirmed)	2.7
06-11	88	Aortic stenosis	1.5
06-11	90	Atresia	1.5
06-11	78	Omphalocele	1.2
06-11	89	Pulmonary stenosis	1.2

Fig 1. Pesticide distribution, weighted by annual number of births in the 2.6km² Section, in the San Joaquin Valley, CA in 1997-2011. Total annual pesticide active ingredients follows a heavily skewed distribution, with almost 95% of observations experiencing less than 5,000 kg per year, and the extreme right tail extending to substantially larger exposures. A small number of observations that exceeded 100,000 kg (annual total in PLS Section) were omitted from this figure for visual clarity. These omissions represented 76 births in multiple counties and years.

REVIEWERS' COMMENTS:

Reviewer #1 (Remarks to the Author):

The authors have addressed all of my comments and the manuscript is suitable for publication.

Reviewer #2 confidentially stated their satisfaction with the revised manuscript

Reviewer #3 (Remarks to the Author):

Well done!